# Anticancer Effects of Propolis Extracts Obtained Using the Cold Separation Method on Breast Cancer Cell Lines

**DOI:** 10.3390/plants12040884

**Published:** 2023-02-16

**Authors:** Marek Gogacz, Jerzy Peszke, Dorota Natorska-Chomicka, Monika Ruszała, Katarzyna Dos Santos Szewczyk

**Affiliations:** 1Chair and Department of Gynecology, Medical University of Lublin, 20-090 Lublin, Poland; 2Department of Experimental Biotechnology, Decont LLC, 08-500 Ryki, Poland; 3Chair and Department of Toxicology, Faculty of Pharmacy, Medical University of Lublin, 20-090 Lublin, Poland; 4Chair and Department of Obstetrics and Perinatology, Medical University of Lublin, 20-090 Lublin, Poland; 5Department of Pharmaceutical Botany, Medical University of Lublin, 20-093 Lublin, Poland

**Keywords:** propolis, cold separation, breast cancer

## Abstract

Propolis and its extracts show a wide spectrum of biological activity. Due to the necessity to use high temperatures and high polarity in the eluent, the obtained extracts are depleted of active compounds. The new, cold separation method allows obtaining a qualitatively better product containing a number of chemical compounds absent in extracts obtained using high-temperature methods. The purpose of our study was to evaluate the biological activity of propolis extracts produced with the cold separation method in four female breast cancer cell lines: MDA-MB-231, MDA-MB-468, MCF-7, and T-47D. The results of the breast cancer cell viability were obtained using the MTT test. Propolis extracts at 75 and 80% showed similar cytotoxicity against cancer cells, with the polyphenol fraction 75% being slightly more negative for cells. Propolis extracts at concentrations of 50, 75, and 100 µg/mL significantly reduced cell viability. With the exception of the MDA-MB-231 line, cell viability was also decreased after incubation with a concentration of 25 µg/mL. Our results suggest that propolis extracts obtained with the cold separation method may be considered as promising compounds for the production of health-promoting supplements.

## 1. Introduction

Breast cancer is the most common malignancy in women worldwide. It is estimated that just in 2018, approximately 2.1 million women worldwide were diagnosed with this type of tumor. In that year, over 600,000 women died from the advanced stage of the disease. The incidence of breast cancer is increasing year by year by 3.1%, starting from approximately 641,000 cases in 1980 to over 1.6 million in 2010. Unfortunately, this unfavorable trend is likely to continue, and in addition, we must include the dark number of cases in the statistics, because the epidemiology of advanced breast cancer is not treated as a priority in most countries. Interestingly, the global burden of this type of cancer is increasing in different countries regardless of the income of the population. This is primarily due to population growth and an aging population. Therefore, the conclusions are not optimistic. The number of patients with this disease is also not exactly known, because official data include mainly diagnoses and deaths but not relapses [1,2,3].

Breast cancer detected at an early stage of the disease, without metastases or cancer that has only spread to the axillary lymph nodes, is considered curable in approximately 70–80% of patients. At an advanced stage, with metastases to distant organs, it is considered incurable with currently available therapies. Advanced-stage breast cancer is a disease for which current therapies aim to prolong the life of the patient with acceptable side effects of the therapy in order to maintain or improve quality of life. This is related to the genetic variability of the cells and the heterogeneity of their structures at the molecular level. This is in turn linked to the activation of the human epidermal growth factor receptor HER2, encoded by ERBB2; the overactivation of hormone receptors, including estrogen and progesterone receptors; and BRCA mutation. The genetic variability of the tumor tissues determines, in this case, the choice of optimal therapy, which is in turn correlated with the type of cell lines and the molecular subtype of the tumor. The therapy itself is interdisciplinary. It includes, among others, surgery in combination with radiotherapy against identified localized tumors and systemic therapies. These include hormone therapy for hormone-receptor-positive tumors, chemotherapy, or anti-HER2 therapy—for tumors with positive HER2 receptors present on the cell surface. In addition, bone-improving drugs and poly(ADP-ribose) polymerase inhibitors are used supportively in BRCA mutation carriers. In recent years, immunotherapy has also been introduced into the arsenal of anticancer agents [4,5,6,7,8,9].

The therapeutic concepts towards which modern oncology is moving are aimed at individualizing therapy and treatment based on tumor biology and early response to treatment. Alongside innovation, equal access to the latest therapies remains a challenge.

Cancer tissue develops in a complex microenvironment consisting of several benign cell types and an extracellular matrix that provides mechanical support to the tumor structure. The most abundant type of cells in a tumor are tumor-associated fibroblasts. In addition, leukocytes, including lymphocytes, macrophages, and stromal cells of myeloid origin are found. Most of these cells are involved in the immune response [10]. Breast cancer tumor tissue varies according to molecular subtypes, which in turn are closely related to immunogenicity. The highest degree is observed in tumors with TNBC and HER2 (+) and lower in luminal subtypes A and B [11,12]. In addition, the response to neoadjuvant treatment and prognosis in breast cancer positively affect the number of lymphocytes infiltrating the tumor, which reflects the intensity of the immune response in the tumor [13,14].

The actual number of metastatic breast cancers after primary presentation in sites and/or organs outside the primary breast tumor area and regional nodes (including ipsilateral subclavian and supraclavicular lymph nodes) depends on several factors, including age, screening, quality initial local treatment and access to drugs, and new treatments (such as precision radiotherapy for brain metastases or access to clinical trials) [15]. In Western countries, the proportion of patients with metastatic recurrence is probably 20–30%.

All tumor characteristics leading to breast cancer metastasis are not yet well understood. In addition, although some researchers are trying to find methods and drugs (such as aspirin and metformin) to prevent the recurrence of metastases, the results so far are mostly inconclusive. Therefore, research is conducted on a large scale on therapies based on drugs that target the molecular mechanisms of tumor development and metastasis, improve the efficiency of the immune system, activate lymphocytes from various groups, inhibit signaling pathways responsible for neovascularization and metastasis, as well as improve the quality of life of patients. Among other things, for this reason, we directed our steps towards a substance already known but still hiding many secrets regarding biological properties—bee propolis.

Propolis is a natural substance produced by honeybees (*Apis mellifera*) from juices, resins, and waxes collected from various parts of plants such as leaves, flower buds, and tree bark. The raw propolis is then mixed with beeswax and enzymes produced by bees in the throat [16]. Bees use this natural material to cover holes, seal cracks in the hive, and maintain constant humidity and temperature in the colony. In addition, it is successfully used to defend the colony against pathogenic microorganisms, parasites, and predators [17,18,19,20,21].

Raw propolis at temperatures above 40 °C is soft and sticky. It is characterized by a specific balsamic aroma. Depending on the environment from which the bees obtain the raw material, the palette of colors is wide, from brown through yellow and green to red [17,22].

The biological activity of propolis has been known for thousands of years. Propolis preparations have been successfully used in traditional and folk medicine for the treatment of gastrointestinal diseases (i.e., gastric ulcers and infections), wounds, ulcers, and burns [19,20,23]. Hippocrates used it to heal wounds. In the 17th century, British pharmacopoeias listed propolis as an official medicine [19]. During World War II, propolis was used as an antibacterial and anti-inflammatory agent [24]. An interesting fact is that this material was used as a component of violin varnish by famous violin makers: Stradivari, Amati, and other masters [19].

The chemical composition of propolis is diverse and depends on the geographical origin, environment, climate, plant resources, place of origin, and time in which it was collected by the bees. The specificity of the local flora is the main factor determining the chemical composition of propolis in addition to its biological and pharmacological properties [19,25].

Pure, raw propolis exhibits antibacterial, fungicidal, antioxidant, immunomodulatory, and anti-inflammatory properties [20,21,26,27,28]. For this reason, among others, propolis is added to a wide range of products used in cosmetology or health care, including creams, gels, skin lotions, shampoos, chewing gums, tinctures, throat sprays, cough syrups, lozenges, soaps, toothpastes, or mouthwashes [21,29,30].

In turn, the chemical composition and biological activity of propolis extracts depend on the methodology of the extraction process. The most commonly used method is the method based on washing the components with 70–75% ethanol [31,32]. Propolis extracts are also obtained by washing with solvents such as water, ethyl ether, methanol, hexane, and chloroform; aqueous solutions of polyethylene glycol, propylene glycol, and glycerol; and using vegetable oil [31,33].

The available methods of propolis chemical composition analysis as well as standardization and quality control methods have been described in many publications. According to literature data, over 300 biologically active compounds [29,31,32,34,35] and several hundred others of unknown structure and properties have been identified in propolis samples of different geographical origin. The main chemical groups found in propolis are flavonoids, aliphatic and aromatic acids, phenolic esters, fatty acids, alcohols, terpenes, β-steroids, alkaloids, and organic acids such as cinnamic acid, o-coumaric acid, *p*-coumaric acid, caffeic acid (CA), or caffeic acid phenylethyl ester (CAPE) [16,18,19,30]. In our previous work [36], we reported the composition of the low-temperature extracts used in this study.

The biological activity of propolis is the result of the interaction of various compounds, and the analysis of the biological activity of each compound separately allows studying the molecular mechanisms underlying the pharmacological properties of the preparation [35].

## 2. Results

### Effect of Propolis Extracts Produced Using the Low-Temperature Method on the Activity of Breast Cancer Cells

In our research, two low-temperature propolis extracts (75 and 80%) were tested on four female breast cancer lines: MDA-MB-231, MDA-M8-468, MCF-7, and T-47D, differing in the molecular profiles. For comparison, cells were treated with cisplatin as a reference compound.

MDA-MB-231 is an epithelial human breast cancer cell line that was obtained from the pleural effusion of a 51-year-old Caucasian woman with metastatic breast adenocarcinoma. This line is one of the most commonly used breast cancer cell lines in breast cancer medical research. MDA-MB-231 is a highly aggressive, invasive, and poorly differentiated triple-negative breast cancer (TNBC) cell line as it lacks estrogen (ER), progesterone (PR), and HER2 (human epidermal growth factor receptor) expression. MDA-MB-468 is an epithelial line isolated from the pleural effusion of a 51-year-old Black patient with metastatic breast adenocarcinoma. MDA-MB-468 is used, among others, to test chemotherapeutic agents and methods of treating breast cancer. MCF-7 is a breast cancer cell line isolated from a pleural effusion in 1970 from a 69-year-old Caucasian woman. It is characterized by the presence of estrogen receptors on its surface, which in turn implies a proliferative response to estrogens. In addition, progesterone receptors are present on the surface of these cells. However, it does not show amplification of the ERBB2 gene (overexpressing the Her2 protein). T-47D is a cell line isolated from a pleural effusion from a 54-year-old patient with infiltrating ductal breast cancer. T-47D cells differ from other human breast cancer cells in that their progesterone receptors are not regulated by estradiol. T-47D cells are used in studies on the effects of progesterone on breast cancer cells and the effects of drugs on transcription regulation.

Human breast cancer cells were treated with serial dilutions (1, 10, 25, 50, 75, and 100 µg/mL) of 80 and 75% propolis extracts for 24 h to investigate the potential cytotoxic effect on this type of cancer.

In our previous study [36], we proved that the tested propolis extracts significantly reduced the viability and inhibited the proliferation of prostate cancer cells (PC-3 and DU-145 cell lines). In the aforementioned work, the results of the influence of both extracts on the physiological line of human fibroblast cells were also presented.

The results of the breast cancer cell viability were obtained using the MTT test (Figure 1, Figure 2, Figure 3 and Figure 4). Both extracts showed similar cytotoxicity against cancer cells, with the polyphenol fraction 75% being slightly more negative for cells. Propolis extracts at concentrations of 50, 75, and 100 µg/mL significantly reduced cell viability. With the exception of the MDA-MB-231 line, cell viability was also decreased after incubation with a concentration of 25 µg/mL.

The identified compounds in the propolis extract produced by Decont belong to various classes in terms of properties against human cell receptors [36]. The selected compounds identified in the extract that show biological activity towards estrogen receptors are presented in Table 1.

Betulin is one of the main compounds identified in this extract. It was found that betulin and its derivatives inhibit tumor growth and cell migration and lead to cell cycle arrest and apoptosis [37]. Moreover, these compounds have been shown to be less toxic towards normal cells than tumor cells [38]. Bache et al. [39] showed strong betulinic-acid-induced cytotoxicity and an early increase in apoptosis in human breast cancer cells under hypoxia. Other compounds identified in our propolis extracts, such as lup-20(29)-en-3-one, 4-tert-octylphenol, and dodecanoic acid, also have a proven cytotoxic effect on various cancer cells [40,41] and they may be responsible for the cytotoxicity of propolis extracts against breast cancer cells.

Propolis contains a wide range of compounds with anticancer activity. These compounds affect, e.g., cell proliferation, the activation of proapoptotic pathways, the modulation of angiogenesis, the modulation of the immune system, or the spreading of metastases. In addition, propolis components affect the structure of the tumor microenvironment and have a chemosensitizing effect on cytostatics. It has been shown that some of the compounds contained in it, such as polyphenols, have an antioxidant and protective effect on normal cells in patients undergoing chemotherapy, thus reducing the side effects of aggressive chemotherapy or radiotherapy and, at the same time, improving the quality of life of patients. In turn, the mechanisms of cytotoxic action of anticancer compounds are most often associated with the induction of apoptosis or disruption of the cell cycle and the inhibition of metastases [16,36].

In order to trace the effect of propolis extracts on cells, it is necessary to understand the mechanism of cell division and, at the same time, their proliferation. In eukaryotes, the stages of the cell cycle are divided into two main phases: interphase and mitosis (M phase). During interphase, the cell grows and copies its genetic material. During the M phase, a cell divides its cytoplasm and DNA into two sets, forming the basis for two new cells. Preparation for division takes place in three phases: G1, S, and G2. Collectively, the G1, S, and G2 phases are known as interphase. Cells in the G1 phase may enter a quiescent state called G0 before engaging in DNA replication.

The life cycle and mitosis are controlled by several mechanisms. The proteins, collectively known as cyclins, regulate the G1, S, and G2 phases by binding and activating cyclin-dependent kinases (Cdks). The phosphorylation of specific sites in regulatory proteins by cyclin-Cdk complexes triggers processes that in turn activate the cell cycle [42,43].

Some of the published data only indicate the effect of propolis and its components on the inhibition of the proliferation process without penetrating the molecular mechanism [44,45,46,47,48,49]. Nevertheless, there are also studies indicating that the substances contained in it modulate the regulators of the cell cycle. For example, cyclin D, Cdk-2/4/6 cyclin-dependent kinases, and cyclin-dependent kinase inhibitors have been identified. Thus, they inhibit cell cycle progression in tumors at the G2/M stage, as well as at the G0/G1 stage, by increasing the expression of p21 and p27 [48,50,51].

The literature data also indicate that some of the substances contained in the extracts induce S-phase cell cycle arrest in MDA-MB-23 breast cancer cells, depending on the dose and duration of action. A dose-dependent decrease in the activity of the G0/G1 phase (CAPE) and elimination of the G2/M phase (CAPE) have also been observed in some types of cancer [52]. The mechanisms of action of biologically active compounds are very different, for example: caffeic acid phenylethyl ester inhibits the ribosomal protein kinase S6 beta-1 (p70S6K), which mediates protein synthesis in the PI3K/AKT pathway and some AKT signaling networks. Its activity was confirmed, among others, against the prostate cancer cells LNCaP, DU-145, and PC-3 [18]. On the other hand, the extract contains artepillin C (ArtC) which attaches to mortalin-p53 complexes, causing the activation of the p53 protein and arresting the growth of cancer cells such as HT1080 (human fibrosarcoma), A549 (human lung cancer), and U2OS (human osteosarcoma) [53]. The inhibitory activity of propolis extracts on the proliferation of HEp2 human epithelial cells through the Stat3/Plk1 pathway by inducing S-phase arrest was also confirmed [54].

Genistein, another component of propolis, inhibits the cell cycle in the G2/M phases. This is achieved by reducing the expression of cyclin B and inducing p21 in a p53-independent manner, as shown by studies on prostate cancer cells [18,36].

Among others, the specific Claudin-2 protein, in turn, is involved in neoplastic proliferation and is highly expressed in human lung adenocarcinoma cells. The expression of this protein is regulated at the transcriptional and post-translational stages. Claudin-2 transcriptional activity is reduced by the inhibition of mitogen-activated protein kinase (MAPKK)/extracellular signal-regulated kinase (ERK)/c-Fos, and phosphatidylinositol-3 kinase (PI3K)/Akt/nuclear factor-κB (NF-κB pathways) [55,56]. Substances contained in propolis extracts reduce the expression of Claudin-2 by reducing the level of p-NF-κB and increasing the level of IκB (NF-κB inhibitor). The inhibition of NF-κB may in turn be involved in the downregulation of Claudin-2 mRNA [57].

Propolis ethanol extract from Turkey was confirmed to induce cell cycle arrest in breast cancer and gastric cancer cell lines MCF-7 and HGC27. Its mechanism of action consists of arresting the cell cycle in the G1/S phase as well as increasing the rate of expression of cell cycle checkpoint proteins. Responsible for this phenomenon are compounds present in the extract, namely, 3-O-methyl quercetin, chrysin, caffeic acid, CAPE, galangin, and pinocembrin. Studies on MCF-7, HGC27, and A549 tumor cell lines have shown that this extract causes cell cycle arrest in the G0/G1 phase by activating p21 [58].

A very important pillar of carcinogenesis is the proinflammatory microenvironment. TLR4 is a protein in the family of Toll-like receptors involved in innate immunity. Aberrant expression of TLR4 has been observed in many types of cancer. An overactivity of this receptor may induce chronic inflammation in the tumor microenvironment, stimulate proliferation, and suppress the apoptosis of cancer cells [59]. The results of studies on the effect of substances present in propolis extracts indicate that one of the pathways inhibiting the proliferation of breast cancers is the inhibition of the Toll-like receptor 4 (TLR4) signaling pathway [60]. With regard to esophageal cancer cells, it has been shown that activation of TLR4 stimulates cell proliferation through the TLR4-MyD88-TRAF6-NF-κB signaling pathway, and the inhibition of NF-κB leads to the inhibition of proliferation [61].

Among others, one of the isoflavonoids present in Brazilian red propolis, vestitol, reduces the level of one of the alpha-tubulin proteins in cells, tubulin in microtubules, and histones H3. Alpha-tubulin and tubulin in microtubules are the proteins responsible for pulling the daughter chromosomes apart during mitosis. The disruption of the microtubule structure in mitosis directly affects cell cycle progression.

Many researchers have discovered that flavonoids and their glycosidic and/or sterol derivatives are able to inhibit cancer cell proliferation and delay tumor progression [62,63]. This mechanism is based on the inhibition of metastasis, the inhibition of angiogenesis [64], and the regulation of some signaling pathways related to apoptosis, such as the Akt and PTEN pathways [65]. This is one of the reasons why introducing foods containing flavonoids into the diet may help prevent the initiation or early progression of cancer cells in cancer patients. One such compound is eupatorin (3′,5-dihydroxy-4′,6,7-trimethoxyflavone), which is one of the candidates for drugs against breast cancer [66]. Previous studies have shown that eupatorin strongly inhibits proliferation and induces apoptosis in many cancer cell lines [67,68].

To sum up, propolis extract can affect a number of signaling pathways in different types of neoplastic cells. It may also affect the tumor microenvironment, making it unfavorable for cell proliferation. In addition, the modulation of the immune system changes its activity against cancerous cells. The whole of these processes, together with their sensitizing effect on the action of chemotherapeutics, gives a general view of a comprehensive, wide spectrum of impacts of these compounds on cancerous tissue.

## 3. Materials and Methods

### 3.1. Reagents

All reagents and solvents used in the experiment were of the highest analytical grade and were obtained from various commercial suppliers. Thiazolyl blue tetrazolium bromide (MTT) and cisplatin were purchased from Sigma-Aldrich, Steinheim, Germany; phosphate-buffered saline (DPBS) from PAN-Biotech GmbH, Aidenbach, Germany; and dimethyl sulfoxide (DMSO) from POCH S.A. Avantor Performance Materials, Inc., Gliwice, Poland.

### 3.2. Propolis Extracts

The raw material was received from various places located in Poland. The material was free of substances derived from the industry and used in plant protection.

A detailed description of the preparation of extracts has been described in our previous study [36].

### 3.3. GC-MS Analysis

The analyses were carried out using an Agilent 8890 gas chromatograph with an Agilent 7010B mass detector and a Gerstel MPS Robotic injector. The parameters of the analysis method were as follows: column: Agilent 19091S-433UI HP-5ms Ultra Inert, 30 m × 250 μm × 0.25 μm, in constant carrier gas (He) mode of 2 mL/min; inlet chamber: MM Inlet with Agilent 5188–6576 cartridge operating at 1:10 sample split at 250 °C; dispensed sample volume: 1 µL; column temperature program: 60 °C (2 min)–10 °C/min–300 °C (10 min); the injector was working at a temperature of 320 °C; the ion source temperature: 280 °C; temperature of mass analyzers: 150 °C; the mass spectrometer was operated in EI mode at 70 eV; and recording mode: spectrum sweep m/z 40–800. The processing of the obtained results was carried out in the program Agilent MassHunter Workstation version 10.1.

### 3.4. Cell Cultures

The four female breast cancer cell lines (MDA-MB-231 (catalog no. HTB-26™), MDA-MB-468 (catalog no. HTB-132™), MCF-7 (catalog no. HTB-22™), and T-47D (catalog no. HTB-133™)) were obtained from American Type Culture Collection (ATCC; Manassas, VA, USA). The MDA-MB-231 and MDA-MB-468 cell lines were cultured in DMEM medium (Dulbecco’s Modified Eagle’s Medium), the MCF-7 cell line in EMEM medium (Eagle’s Minimal Essential Medium), and the T-47D cell line in RPMI-1640 medium supplemented with 10% FBS (fetal bovine serum) and antibiotics: 100 U/mL of penicillin, 100 μg/mL of streptomycin, and 2.5 μg/mL of amphotericin B (all media and drugs were purchased from PAN-Biotech GmbH, Aidenbach, Germany). Cells were grown in a humidified incubator at 37 °C and 5% CO_2_ atmosphere in 75 cm^2^ tissue culture flasks.

### 3.5. Cell Cytotoxicity Assay

The inhibitory effects of both propolis extracts on breast cancer cell growth were assessed using the MTT assay (European Centre for the Validation of Alternative Methods, Database Service on Alternative Methods to Animal Experimentation). Cell viability was determined using a mitochondria-dependent reaction (reduction in mitochondrial dehydrogenase activity) based on the measurement of formazan production from the MTT salt and was expressed as the percentage of viable control cells. After evaporating the residual ethanol and drying the preparations at room temperature, the weighted samples of the extracts were dissolved in DMSO to obtain a stock solution and subsequently diluted to the required concentration with the appropriate cell culture medium. The series of solutions were prepared *ex tempore*. The cells at a density of 1 × 10^5^ cells/mL in 96-well plates were exposed to six different concentrations (1, 10, 25, 50, 75, and 100 µg/mL) of the tested extracts for 24 h at 37 °C. Afterwards, the incubation mixture was removed from each well of a microplate and 200 µL of MTT solution (5 mg/mL) was added. After incubation for 3 h at 37 °C, the solution was removed carefully from each well, and 100 µL of DMSO was added to dissolve the formazan crystals. The absorbance was measured at 550 nm using a Power Wave microplate reader (BioTek Instruments, Inc., Winooski, VT, USA). Based on the MTT assay results, the IC_50_ values for the tested extracts were derived from the concentration–response curves. The DMSO concentration in the incubation mixture did not exceed 0.1% *v*/*v*.

### 3.6. Statistical Analysis

The results obtained from the MTT assay were analyzed statistically using the GraphPad application (GraphPad v.5.01, GraphPad Software, Inc., La Jolla, CA, USA). Data are presented as mean ± standard deviations (± SD). Statistical analyses were performed using one-way ANOVA with Tukey’s post hoc test. *p* < 0.05 was considered to indicate a statistically significant difference.

## 4. Conclusions

Propolis extracts contain a number of compounds that affect cell signaling pathways. These compounds have a multidirectional effect on the proliferative mechanisms in cells, both at the level of gene expression and regulatory mechanisms. Propolis extracts obtained using the low-temperature method contain compounds that have a cytotoxic effect on breast tumor cells, regardless of the degree of expression of estrogen receptors on the surface of these cells. The results of our research indicate that even at a concentration of 50 micrograms/mL of propolis extract, signaling pathways in cells are damaged, which leads to the inhibition of metabolic activity in them. However, further in vivo clinical and animal studies of the tested extracts are required to evaluate their modes of action and potential side effects.

## Figures and Tables

**Figure 1 plants-12-00884-f001:**
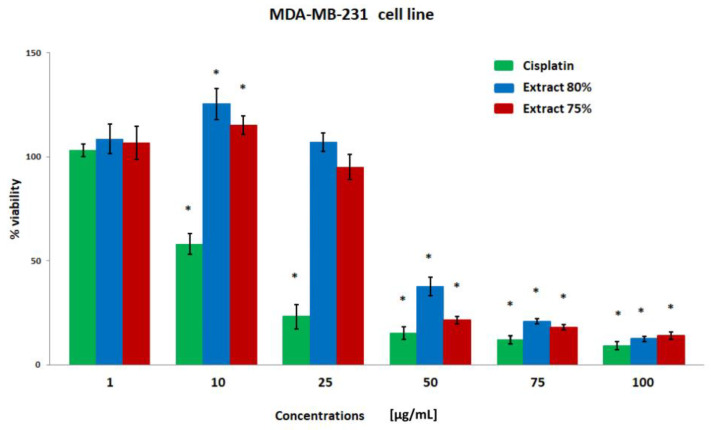
The viability of the MDA-MB-231 cell line (%) after 24 h incubation with various concentrations of cisplatin, extract 80%, and extract 75% measured with MTT test.. Significant values (*) compared with Cisplatin with *p* < 0.05 (one-way ANOVA with Tukey’s post hoc test).

**Figure 2 plants-12-00884-f002:**
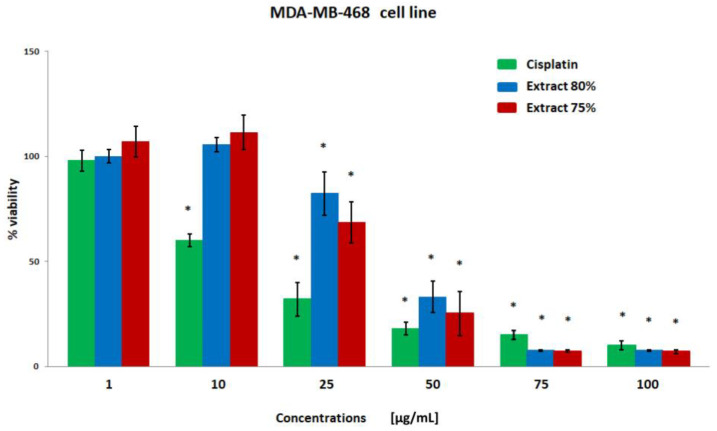
The viability of the MDA-MB-468 cell line (%) after 24 h incubation with various concentrations of cisplatin, extract 80% and extract 75% measured with MTT test. Significant values (*) compared with Cisplatin with *p* < 0.05 (one-way ANOVA with Tukey’s post hoc test).

**Figure 3 plants-12-00884-f003:**
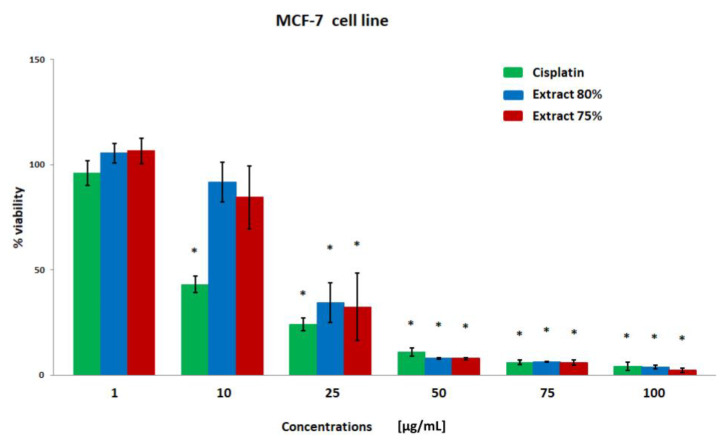
The viability of the MCF-7 cell line (%) after 24 h incubation with various concentrations of cisplatin, extract 80%, and extract 75% measured with MTT test. Significant values (*) compared with Cisplatin with *p* < 0.05 (one-way ANOVA with Tukey’s post hoc test).

**Figure 4 plants-12-00884-f004:**
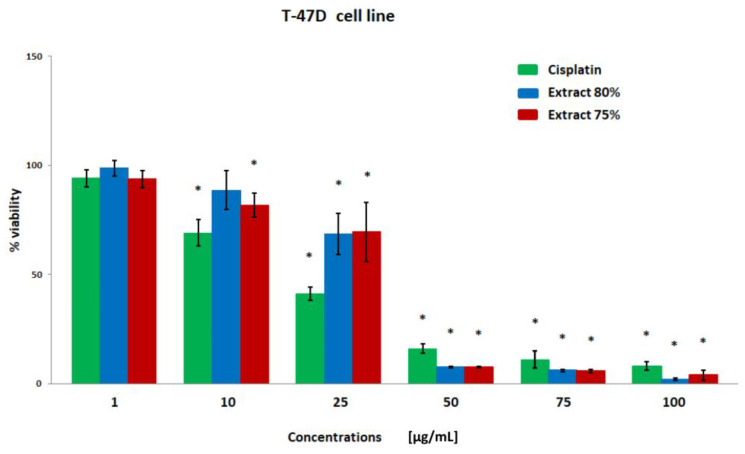
The viability of the T-47D cell line (%) after 24 h incubation with various concentrations of cisplatin, extract 80%, and extract 75% measured with MTT test. Significant values (*) compared with Cisplatin with *p* < 0.05 (one-way ANOVA with Tukey’s post hoc test).

**Table 1 plants-12-00884-t001:** Selected compounds identified in the low-temperature propolis extract that show biological activity towards estrogen receptors.

Compound	CAS
[1,3,5]Triazine-2,4-diamine, 6-(imidazol-1-yl)-N,N′-di(p-tolyl)-	1000304-62-6
1H-Pyrrolo[2,3-b]pyridine, 3-(1-piperidinylmethyl)-	23616-64-0
2,2′-Bis(4,5-dimethylimidazole)	69286-06-2
Acenaphtho[1,2-b]quinoxaline, 9-methoxy-	26832-43-9
Benzo[g][1]benzothiopyrano[4,3-b]indole	10023-23-1
Betulin	473-98-3
Carbamodithioic acid, dimethyl-, 2,3,5,6-tetrachloro-4-pyridinyl ester	1000305-31-1
Dodecanoic acid, ethyl ester	106-33-2
Lup-20(29)-en-3-one	1617-70-5
Lup-20(29)-en-3-ol, acetate, (3.beta.)-	1617-68-1
Cinnamic acid, 3,4-dimethoxy-, trimethylsilyl ester	27750-71-6
dl-7-Azatryptophan	1137-00-4
Phosphine, dicyclohexyl[1,2-di(2-pyridyl)ethyl]-	1000158-19-7
(-)-Neoclovene-(I), dihydro-	1000152-82-1
1H-Cycloprop[e]azulene, decahydro-1,1,7-trimethyl-4-methylene-	72747-25-2
1H-Pyrrole, 2,4-diphenyl-	3274-56-4
2-(1,1-Dimethylethyl)-6-(1-methylethyl)phenol	22791-95-3
2-Naphthaleneacetonitrile, 6-methoxy-.alpha.-methyl-	86603-94-3
4-Hydroxyphenylethanol	501-94-0
9,19-Cyclolanost-24-en-3-ol, acetate, (3.beta.)-	1259-10-5
9H-Carbazole, 9-methyl-	1484-12-4
14,17-Nor-3,21-dioxo-.beta.-amyrin, 17,18-didehydro-3-dehydroxy-	1000132-26-8
Acenaphthene	83-32-9
Androstan-17-one, 3-[(triethylsilyl)oxy]-, (3.alpha.,5.alpha.)-	65598-66-5
Chrysin	480-40-0
Dibenz[a,h]anthracene, 5,12-diphenyl-	14474-66-9
Pyrazolo[1,5-a]pyridine, 3-methyl-2-phenyl-	17408-32-1
Quinoline, 2-phenyl-	612-96-4
Trimethyl[4-(1,1,3,3,-tetramethylbutyl)phenoxy]silane	78721-87-6

## Data Availability

Not applicable.

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
