# Peer review of "Anticancer Effects of Propolis Extracts Obtained Using the Cold Separation Method on Breast Cancer Cell Lines"

_plants, 2023, doi:10.3390/plants12040884_

Round 1

Reviewer 1 Report

The manuscript by Gogacz et al. describes the study of anticancer properties of propolis extracts obtained by cold separation method utilized by the authors earlier.

Introduction describes the peculiarities of breast cancer proliferation, metastases, etc. This introductory description is useful but too long and should be shortened.

Lines 152-198 of Results section are suitable rather for Introduction than for Results.

In Results section, the authors give just a statement about the influence of their previously obtained extracts on the newly observed cell viabilities (Figures 1-4).  However,  these results were not properly discussed in terms of concentrations, composition, and other properties of the investigated extracts. Instead, in the text starting from Page 7, the authors  present a kind of literature review concerning the effects of different substances and mechanisms of these effects described in earlier publications.

I recommend major revision of the manuscript before publication. A deeper analysis of the new obtained results should be included.

Reviewer 2 Report

The manuscript entitled Anticancer Effects of Propolis Extracts Obtained by the Cold Separation Method on Breast Cancer Cell Lines is interesting and easy to read. The manuscript promises a very interesting topic with the title and abstract, but the introduction is too focused on describing the incidence of breast cancer and this part must be shortened. Introduction does not state the novelty. By citing statistics and already known facts about breast cancer, there is a lack of already conducted research and confirmed hypotheses regarding the effect of natural materials on cancer cells and, consequently, what kind of research it is. Thus, part of the text in the results (lines 152-179) should be used in the introduction. Due to the excessively long introduction, which does not provide a satisfactory novelty to the otherwise interesting topic of the manuscript, the number of references used is large and it is difficult to follow them all. The research on cell lines is satisfactory and appropriate, as are the comments and methods described.

According to the comment, I suggest the authors to edit the introduction and results text as suggested, focused on expression the novelty and importance of the conducted research.

Round 2

Reviewer 1 Report

In the revised version of the manuscript, the authors have significantly improved the manuscript. It can be accepted for publication.

Author Response

Dear Reviewer,

We are thankful to the reviewer for reviewing and giving valuable suggestions to further improve our manuscript. We have carefully gone over the varies queries raised and have tried to address them. We hope that the reviewer will be satisfied with the changes that have been incorporated. All changes in the text are highlighted in green.

Sincerely Yours,

 Authors